# Primary and Secondary Cone Cell Death Mechanisms in Inherited Retinal Diseases and Potential Treatment Options

**DOI:** 10.3390/ijms23020726

**Published:** 2022-01-10

**Authors:** Alicia A. Brunet, Alan R. Harvey, Livia S. Carvalho

**Affiliations:** 1Centre for Ophthalmology and Visual Sciences, The University of Western Australia, 35 Stirling Hwy, Crawley, WA 6009, Australia; liviacarvalho@lei.org.au; 2Lions Eye Institute Ltd., 2 Verdun St, Nedlands, WA 6009, Australia; 3School of Human Sciences, The University of Western Australia, 35 Stirling Hwy, Crawley, WA 6009, Australia; alan.harvey@uwa.edu.au; 4Perron Institute for Neurological and Translational Science, 8 Verdun St, Nedlands, WA 6009, Australia

**Keywords:** inherited retinal diseases, cell death, oxidative stress, apoptosis, necroptosis, autophagy, immunological effects, epigenetic, treatment

## Abstract

Inherited retinal diseases (IRDs) are a leading cause of blindness. To date, 260 disease-causing genes have been identified, but there is currently a lack of available and effective treatment options. Cone photoreceptors are responsible for daylight vision but are highly susceptible to disease progression, the loss of cone-mediated vision having the highest impact on the quality of life of IRD patients. Cone degeneration can occur either directly via mutations in cone-specific genes (primary cone death), or indirectly via the primary degeneration of rods followed by subsequent degeneration of cones (secondary cone death). How cones degenerate as a result of pathological mutations remains unclear, hindering the development of effective therapies for IRDs. This review aims to highlight similarities and differences between primary and secondary cone cell death in inherited retinal diseases in order to better define cone death mechanisms and further identify potential treatment options.

## 1. Introduction

Inherited retinal diseases (IRDs) are a class of disorders that cause visual dysfunction, with currently more than 260 disease-causing genes identified [1]. IRDs are the leading cause of blindness in the working age population, affecting approximately 1:2000 people worldwide [2]. The estimated total worldwide economic burden of all visual impairments to global health systems $USD 3 trillion [3]. Currently, there is only one approved treatment for one IRD disease gene: Luxturna, a gene therapy for the treatment of Leber congenital amaurosis type 2 (LCA2) [4]. Although gene therapy remains a promising primary treatment option, therapies are not being developed at a sufficiently fast rate to meet their increasing demand; only an estimated 13.8% of drugs trialled have been clinically approved [5]. Gene therapies should also be tailored for each individual disease-related gene, contributing massively to treatment expenditure [6].

The majority of IRD mutations affect rod and cone photoreceptors, the cells in the retina primarily responsible for light detection. Cones are responsible for daylight (photopic) vision, colour vision, and visual acuity; thus, degeneration of cone photoreceptors has the largest impact on the quality of life for people with IRDs [7]. Rod photoreceptors mediate vision during dim-light (scotopic) conditions and are important for night vision, but they lack colour sensitivity [8]. Interestingly, cones are remarkably sensitive to degeneration in IRD. Their degeneration can occur either directly via mutations in cone-specific genes (primary cone death), as seen in conditions such as achromatopsia or cone dystrophies, or indirectly as in conditions such as retinitis pigmentosa, where a primary degeneration of rods is followed by a secondary degeneration of cones (secondary cone death) [9,10]. Congenital achromatopsia, or total colour blindness, is a rare autosomal recessive cone-dystrophy affecting 1:30,000 people worldwide [1,11]. It results in the loss of colour discrimination, fine visual acuity, and photophobia [11]. Retinitis pigmentosa is a debilitating eye disease with approximately 60 disease-associated genes [1] and affects 1:4000 people [12]. Retinitis pigmentosa makes up 40% of IRD cases, and there are over one million individuals living with the disease worldwide [12]. It is caused by the primary degeneration of rod photoreceptors, which causes night blindness, generally occurring in young adulthood. Rod degeneration is followed by the secondary degeneration of cone photoreceptors, resulting in progressive blindness starting from the periphery, then advancing centrally until vision may be completely lost in later years [12]. A full description of the clinical and genetic characterisation of the different IRDs can be found elsewhere [13].

Despite the crucial role of photoreceptors in vision, the cell death mechanisms linked to different IRD mutations have not been fully elucidated. Identifying biological changes occurring in diseased photoreceptors—particularly the cones—will help advance the treatment of IRDs. Moreover, finding a common treatment that is effective for both primary and secondary cone degeneration will lessen the economic burden of developing individual therapies for each disease mutation. Current research uses animal models to investigate certain treatments and their efficacy.

Animal models are a valuable tool to study biological changes in the retina as they allow an in-depth analysis of functional, structural and molecular changes of the retina in response to photoreceptor degeneration. Mice in particular develop a similar pathophysiology to humans, and there are over a hundred mouse models of retinal degenerations currently available to researchers [14]. Achromatopsia is a cone-dystrophy commonly studied, with many studies investigating the *Cnga3^−/−^*, *Cngb3^−/−^* models which make up approximately 30% and 50% of achromatopsia cases, respectively [11], as well as the cone photoreceptor function loss 1 (*Cpfl*) model. Studies into the rod-cone dystrophy retinitis pigmentosa models have a large clinical significance due to mutations making up 40% of IRD cases [12]. The widely studied retinal degeneration 1 (*Rd1*) model, first discovered in 1924 [15], has been the predominant model used to study retinitis pigmentosa. Another model widely used to study retinitis pigmentosa is the *Rd10* model which exhibits slower photoreceptor degeneration compared with the *Rd1*. Collin et al. [16] offer a comprehensive overview on genes and associated IRD mouse models, and Winkler et al. [17] recently reviewed the literature on large animal IRD models. The objective of this current review is to distinguish the similarities and differences between primary and secondary cone degeneration in IRDs by summarising findings from in vivo IRD studies of the last 20 years in the hope that better understanding of cone death mechanisms will result in the discovery and clinical development of novel therapies.

## 2. Mechanisms of Cone Cell Death

Cell death involves complex, often intersecting cellular pathways; thus, for example some researchers now regard apoptosis and necrosis as degenerative events at either end of a cell death spectrum rather than an either/or scenario [18]. Autophagy is another mode of cell death which appears to be more related to apoptosis [18]. Some cell death markers can occur in multiple cell death pathways [19], making identification of one particular or specific pathway in different IRDs difficult. Several studies of cone cell death have been carried out on animal models of primary and secondary cone degeneration; however, there is not always consensus as to the mechanism of photoreceptor degeneration in the different IRD types. This may be due to different animal models being used, different mutations within the animal models, different age-points for investigation, amongst other variables. Earlier studies may have also had a lack of understanding of the cell death spectrum and the potentially different modes of cell death, which in recent years have been better characterised. Below we discuss the different types of cell death mechanisms that have been reported in IRD models in which cone degeneration is present.

### 2.1. Apoptosis

Apoptosis is an active programmed cell death characterised by cell shrinkage, minimal or no inflammatory response, and the formation of apoptotic bodies [18]. It was first described in 1972 by Kerr, Wyllie, and Currie, where they identified apoptosis as being required for maintaining homeostasis in an organism by removing damaged or infected cells, as well as focal elimination of cells during normal embryonic development [20]. Dysregulation of apoptosis can be detrimental and contributes to neurodegenerative diseases such as Alzheimer’s disease [21], Parkinson’s disease [22], and Huntington’s disease [23]. In IRDs, there is conflicting literature about whether photoreceptors are degenerating by apoptosis in both primary and secondary cone degeneration models. As apoptosis is one of the earliest identified cell death mechanisms, more research has been done on apoptosis markers in IRDs than other cell death mechanisms. Earlier studies reported that photoreceptors in IRDs were dying through apoptosis based on the presence of in situ terminal deoxynucleotidyl transferase dUTP nick-end labelling (TUNEL). However, at least in some tissues, this process has been reported to label not only apoptotic but also other forms of cell death, including necrosis [24]. It is important to note that certain apoptosis-related markers are also linked to other cell death mechanisms such as necroptosis and autophagy [18]. For example, the extrinsic apoptosis pathway shows various similarities to the necroptosis programmed cell death pathway. Another possible contributing factor to variability in results is the temporal progression of disease in IRD models, where certain death-associated molecules may be expressed at different times in different IRD models. This section discusses major apoptosis hallmark features, as well as atypical apoptosis pathways that have been investigated in IRDs, and whether these pathways differ between primary and secondary cone degeneration.

A typical hallmark feature researchers search for when investigating apoptosis is increase in caspase activity, as caspases are an important molecule in classical apoptosis pathways. Caspases are a family of protease enzymes that can be involved in both intrinsic and extrinsic apoptosis. Intrinsic activation occurs during cell stress, where intracellular signals from the mitochondria initiate a cell death cascade (Figure 1). This occurs through cytochrome c leakage from the mitochondria after pro-survival inhibition of B-cell lymphoma 2 (BCL2), and binding of proapoptotic proteins BCL2-like protein 4 (BAX) and BCL2 homologous antagonist killer (BAK) to the mitochondria. Release of cytochrome c leads to oligomerisation of apoptotic protease activating factor-1 (APAF1), which is involved in the formation of the apoptosome complex, causing activation of initiator caspase-9, and executioner caspase-3 and -7. This final caspase activation results in apoptosis of the cell.

Extrinsic apoptosis also involves caspases and occurs due to the binding of extracellular ligands to surface receptors of the cell (Figure 1). Cell death receptor agonists, such as Fas ligand (FasL), tumour necrosis factor-related apoptosis inducing ligand (TRAIL), or tumour necrosis factor α (TNFα), bind to their corresponding receptors to initiate apoptosis. The binding of FasL to the Fas receptor or TRAIL to the TRAIL receptor recruits Fas-associated protein with death domain (FADD) and procaspase-8 to form the death inducing signalling complex (DISC), activating the initiator caspase-8 to activate the caspase cascade involving caspase-3, -6, and -7, resulting in cell apoptosis. Binding of TNFα to the TNF receptor 1 (TNFR1) leads to the formation of complex I, which consists of TNFR1-associated death domain (TRADD), receptor-interacting serine/threonine-protein kinase 1 (RIP1), TNFR1-associated factor 2 (TRAF2), and cellular inhibitor of apoptosis protein (cIAP). Dissociation of RIP1 or cIAP from complex I can lead to TRADD and RIP1 binding to FADD and procaspase-8 to form complex II. Formation of complex II leads to the cleavage of procaspase-8 to caspase-8, following the same caspase activation cascade as FasL apoptosis. However, complex II formation may also lead to necroptosis with the addition of receptor-interacting serine/threonine-protein kinase 3 (RIP3), an important necroptosis marker. Dysregulation of FADD can also be found in autophagy [25,26], whilst the FasL and TRAIL pathways may also cause necroptosis. Therefore, dysregulation of markers involved in extrinsic apoptosis, with the exception of caspase-8 which blocks autophagy and necroptosis activity, cannot necessarily give an indication of the mechanisms that are associated with this type of cell death.

The classic apoptosis pathway involving caspase activation has been found in both primary and secondary cone degeneration IRD models, but their activation in IRD models is still largely disputed due to differing findings in the literature. During primary cone degeneration, caspase-3 activation was found in the *Cnga3^−/−^* mouse model [27], a widely used model of achromatopsia, a type of primary cone degeneration IRD. Other activated caspases such as caspase-7 and caspase-12 have also been identified in the *Cnga3^−/−^* retina as well as in the *Cngb3^−/−^* mouse model (another model of achromatopsia), suggesting apoptosis is present in primary cone death [28,29]. Power et al. [30] investigated caspase-3 in the *Cpfl1* mouse of primary cone death and in the *Rd1* and *Rd10* mice of retinitis pigmentosa (secondary cone death), showing age-dependent caspase-3 activation in all three models [30]. Further studies into the secondary cone degeneration in the *Rd1* mouse have shown caspase-3 activation as well as caspase-7, -8, and -12, although they do not differentiate whether this is occurring in rods or cones [31,32,33,34,35,36]. However, another study by Arango-Gonzalez and colleagues was not able to replicate these classical apoptosis marker findings in 10 different murine lines of both primary and secondary degeneration [37]. This study identified the peak in photoreceptor death in each IRD model and at that point immunostained for caspase-3 and caspase-9. The only model that was positive for caspase activity was the *Rho* S334ter rat model for retinitis pigmentosa, contradicting previous findings of caspase activity. These findings also implicate that different IRD mutations may lead to differences in photoreceptor cell death. The lack of caspase activation was also evident in other studies of IRDs [38,39]. Furthermore, studies inhibiting the caspase cascade failed to promote photoreceptor survival, contributing to the argument that photoreceptor degeneration in IRDs is caspase-independent [38,40]. Caspase activation has been a molecular marker widely used to identify whether cells are undergoing apoptosis or apoptotic-like cell death, but it is still unclear whether caspase activation is prevalent in IRDs and the current data would seem to suggest that caspase inhibition may not be a feasible treatment approach.

The classic intrinsic apoptosis pathway is the more understood pathway of cell death, and has been more widely studied in secondary cone degeneration models such as the *Rd1* mouse due to their earlier discovery/availability compared twith primary cone degeneration models. Bax activation and cytochrome c leakage was found in three different retinitis pigmentosa models, including the *Rd1* model, but the studies did not discriminate whether activation was in rods or cones [31,41]. Increased BAX activation was also found in the *Rpe65^−/−^* mouse for Leber congenital amaurosis; however, inhibition of BAX only slowed rod, but not cone, degeneration [42]. Furthermore, the Arango-Gonzalez team was unable to detect activity of BAX in 9 out of 10 IRD mouse lines of primary and secondary cone degeneration that were investigated [37]. Promotion of the BCL2 anti-apoptotic cell survival pathway to preserve photoreceptors in secondary cone degeneration mouse lines has also failed [43], and cytochrome c leakage was also not detected in certain IRDs models [37,39,44]. However, the absence of hallmark apoptosis marker activation is not necessarily an indication of absence of apoptosis, as the extrinsic apoptosis pathway and other apoptosis pathways are still largely unexplored in IRDs.

Calpains are a family of Ca^2+^-dependent cysteine proteases that are involved with atypical apoptosis pathways. Calpains are also associated with other forms of cell death such as autophagy and necroptosis. Calpain 1 and 2 are expressed in the retina and both are associated with endoplasmic reticulum (ER) stress-induced apoptosis. Calpain 1 induces apoptosis by cleaving apoptosis-inducing factor (AIF) in the mitochondria to form truncated AIF (tAIF), which translocates to the cell nucleus to initiate DNA fragmentation and cell death [45]. Calpain 2 cleaves procaspase-12 located in the endoplasmic reticulum to produce activated caspase-12, while also deactivating the anti-apoptotic molecule B-cell lymphoma-extra large (BCL-XL) to induce apoptosis. Previous studies have implicated calpain activity in contributing to photoreceptor degeneration in IRDs, and there is evidence from both primary and secondary cone degeneration models that calpain is activated in cone degeneration [29,32,33,35,37,46,47,48]. Blocking either calpain 1 or 2 is not effective in preventing cone degeneration; however, a combined treatment to block the activity of both calpain 1 and 2 increased photoreceptor survival in both primary and secondary cone degeneration [29,33]. Although this treatment showed only short-term survival effects for cones, repeated treatment of calpain inhibitors may be a potential therapy option for those affected by IRDs.

Studies into primary cone degeneration are underrepresented compared with their secondary cone degeneration counterpart, possibly due to the limited disease genes associated with primary cone degeneration. In humans, there are only six known genes to cause congenital achromatopsia. The *CNGA3* and *CNGB3* genes contribute to approximately 65-80% of achromatopsia cases, and therefore, research has been focused on the *Cnga3^−/−^* and *Cngb3^−/−^* animal models to study primary cone degeneration. This may skew interpretation of primary cone degeneration as more recent papers have suggested a non-apoptotic death mechanism, albeit a proper mechanism has not been fully defined. However, animal models carrying mutations on other achromatopsia or cone dystrophy-associated genes may have different cell death pathways. Therefore, further investigation is required into primary cone degeneration and the different animal models available.

The secondary cone degeneration pathway is generally considered to be more apoptotic in nature. The primary degeneration of rods creates an unstable retinal environment for cones to survive. Rods make up around 95% of the cells in the retina [49]. Their degeneration is theorised to increase oxygen levels leading to oxidative stress [35,50,51,52], decrease in rod-derived support factors [53,54], increase in glial activation leading to chronic immune responses [55,56,57], and release of toxins [58], all potentially contributing to subsequent cone degeneration. In *Rd1* mice, cone loss begins when almost all rods have degenerated, starting with the loss of cone outer segments which become depleted by postnatal day 21 (P21) [59]. In humans, where the great majority of cones are located in the central foveal region of the retina, vision loss in retinitis pigmentosa starts at the periphery and progresses centrally until patients may become fully blind. Rodents lack a cone rich fovea and have a more uniform distribution of cones across the retina [60,61]. Nonetheless, there is an observed spatio-temporal dependent degeneration of different types of cones in the retina demonstrated in the *Rd1* mouse, where S-opsin cones were more resistant to degeneration in the inferior retina whilst M/L-opsin cones were more resistant in the superior retina [59].

### 2.2. Necroptosis

Necrosis occurs due to cellular insult such as physical trauma, infections, or toxins. Necroptosis is a programmed form of necrosis and is associated with cell swelling, cell lysis and inflammation, usually affecting multiple cells at a time [62]. Necrosis also exhibits the same physical disruptions to cell function, making the two cell death processes indistinguishable through histological methods of analysis [63]. However, necrosis is less considered in genetic diseases due to it being more frequently associated with unregulated external damage [64].

Identifying whether cells are undergoing necroptosis is difficult, as many molecular markers involved with necroptosis are also associated with other cell death pathways. RIP1 and RIP3, key factors in necroptosis, form the necrosome complex, which phosphorylates mixed lineage kinase domain-like pseudokinase (MLKL) that eventually leads to eliciting an immune response and necroptotic cell death (Figure 1) [65,66,67]. It is important to distinguish between RIP3 and RIP1, as RIP1 participates in apoptosis as well as necroptosis [65]. MLKL is also associated with apoptosis by eliciting cell death through the protein kinase-like endoplasmic reticulum kinase-eukaryotic initiation factor 2 α subunit (PERK/eIF2α) pathway [68]. A key protein exclusively linked to necroptosis is RIP3, which forms the RIP1-RIP3 necrosome complex [65,66,67]. Extrinsic apoptosis actively suppresses necroptosis through caspase-8, inhibiting the formation of the RIP1-RIP3 necrosome complex [69,70]. Therefore, the presence of RIP3 or caspase-8 may be indicative of necroptosis or apoptosis occurring, respectively. Conversely, as far as current literature pertains, caspase-8 expression in IRDs has not been explored. The RIP3/caspase-8 dichotomy may be an interesting marker in identifying whether cones are degenerating by a more apoptotic or necroptotic mechanism in IRDs.

In primary cone degeneration, a study into the *Pde6c^−/−^* zebrafish model of achromatopsia showed that knockdown of the *RIP3* gene rescued photoreceptor cell death [71]. In secondary cone degeneration, a study on the *Rd10* mouse model of retinitis pigmentosa, found that RIP3 activity mediated cone cell death but not rod cell death [72]. These results were mirrored by Viringipurampeer et al. [73] in the *Rho* S334ter rat model, where RIP3 activation was correlated with cone cell death, whereas rods died by a caspase-dependent apoptosis mechanism. However, in the same study using a different retinitis pigmentosa model, the *Rho* P23H rat, it was found that rods died via a RIP3 necroptotic mechanism, whereas cone loss was attributed to inflammasome activation. Detection of particular apoptosis and necroptosis markers in both gene and protein expression was also age-dependent [73]. Differences in cell death, being both mutation and age-dependent in animal models, means that disease models need to be fully characterised for disease progression and cell death mechanisms before generalised treatments can be translated into potential therapies.

The inclination to investigate necroptotic mechanisms in IRDs may be linked to an inability to conclusively identify certain apoptosis markers and pathways in mouse models, leading to conflicting literature. As stated in the above outline of apoptosis, some studies have found apoptosis markers in primary and secondary cone degeneration, whilst others have been unable to identify such markers. Other studies have attributed cell death markers to being non-apoptotic, such as histone deacetylases (HDACs) and protein kinase G (PKG), and have concluded that cone degeneration in IRDs involves some form of non-apoptotic mechanism [37,74], even though these molecules can also be activated in apoptosis [75,76,77]. Overall, these considerations highlight the need for a better understanding and description of the complex, intersecting cell death pathways that can be involved in photoreceptor loss. Whereas the initial research focused primarily on the classic apoptosis pathway to identify mechanism of cone death in IRDs, it is becoming evident that other atypical apoptosis pathways, as well as other cell death mechanisms, should be explored further.

### 2.3. Autophagy

Autophagy is a relatively newly discovered cellular mechanism that in recent years has been attracting attention [78]. Autophagy is a metabolic regulating mechanism occurring naturally in the body to clear damaged cells and debris, as well as preventing cytotoxic protein build-up. Three types of autophagy have been identified and include (1) macroautophagy, (2) chaperone-mediated autophagy, and (3) microautophagy—with all types moving cellular cargo to the lysosome for their degradation [79,80]. Macroautophagy is the most understood and occurs at low levels during normal conditions as a cellular maintenance mechanism by sequestering cell components—forfor example, damaged or excessive organelles—in autophagosomes for degradation. During cellular stress conditions, such as nutrient deficiency or energy starvation, degradation of cytoplasmic organelles is used to generate additional metabolites to aid cell survival [80].

Autophagy may be important for cone survival, as activation during stressed conditions may support cone metabolic needs [81]. However, an overwhelming increase in autophagic activity can lead to cellular death [82] and could also be a contributing factor to cone cell death in IRDs. A study using the induced light-damaged mouse model of retinal degeneration showed that photoreceptors were dying by autophagy activation, which was potentially eliciting an apoptosis response. Autophagy and apoptosis were blocked separately using specific inhibitors to each pathway, with both instances partially preserving photoreceptors. Interestingly, it was found that blocking both autophagy and apoptosis lead to cell death via a necrotic mechanism [83].

In secondary cone degeneration, due to the initial massive depletion of rods, there is loss of rod-derived support factors [53]. Consequently, cones degenerate in what is theorised to be an autophagic cell death mechanism [54,84]. The rod-derived cone viability factor (RdCVF) is thought to play an important role in trophic support for cone photoreceptors [53,85,86]. RdCVF binds to the Basigin-1 (BSG1) cell surface receptor, which is also bound to the glucose transporter-1 (GLUT1) channel to promote glucose uptake into the cell (Figure 2). When rods degenerate, there is a decrease in glucose uptake into cones leading to a decrease in cone activity [85]. This decrease in glucose also affects the mammalian target of rapamycin (mTOR) pathway resulting in poor nutrient conditions in cones (Figure 2). As a last resort, autophagy is activated to preserve cellular nutrients through reabsorption of intracellular proteins and organelles into vacuoles for recycling [54]. Supplementation of RdCVF in mouse models of retinitis pigmentosa (secondary cone degeneration) was shown to preserve cone photoreceptors [86,87]. Furthermore, stimulation of the mTOR pathway in mouse models of retinitis pigmentosa (*Pde6b^−/−^*, *Pde6g^−/−^ Rho^−/−^* and *Rho* P23H) has been shown to prolong cone survival and may present as a potential treatment for patients with secondary cone degeneration [54]. Pharmacological stimulation of autophagy in the secondary cone degeneration model *Rho* P23H led to accelerated photoreceptor degeneration, whilst inhibition of autophagy preserved photoreceptor structure and function [88]. Targeting the RdCVF pathway to preserve cones in secondary degeneration models may be promising, but whether preservation of cones using RdCVF can be translated for primary cone degenerations is unknown.

Certain autophagy-related (ATG) genes involved with autophagosome formation are also implicated in apoptosis activation, drawing a close relationship between autophagy and apoptosis. The ATG proteins ATG12, ATG5, and ATG16 congregate to initiate the elongation process of autophagosome formation [89]. However, ATG12 may function independent of the ATG12-ATG5-ATG16 complex, where it binds and inactivates the pro-survival gene b-cell lymphoma 2 (BCL2) to promote apoptosis [90]. ATG5 may also function independently from the autophagosome complex. When cleaved by calpains, ATG5 translocates from the cytosol to the mitochondria, where it interacts with BCL2-like 1 (BCL2L1, also known as BCL-XL) to induce cytochrome-c release and caspase activation leading to apoptosis [91]. Interestingly, deletion of ATG5 specifically in mouse rod photoreceptors led to a decrease in autophagy, but also an increase in apoptotic degeneration of rods [92], while in mouse cones ATGG5 depletion resulted in minimal deleterious effects up until 4 months of age [81].

On the other hand, autophagy activation in some instances appears to inhibit apoptosis, serving in a protective role during the disruption of homeostasis [93]. Autophagy has been shown to inhibit the negative effects of apoptosis in a rodent model of retinal detachment [93] as well as in neurodegenerative diseases such as Parkinson’s [94], highlighting the fragile equilibrium between cell survival and cell death-regulated autophagy. However, these beneficial effects of autophagy may not directly translate to prevent photoreceptor degeneration in IRDs. Taken together, the use of potential autophagy therapies in IRDs may prove difficult to implement due to the delicate cell survival/death equilibrium.

## 3. Cellular Stress

### 3.1. Oxidative Stress

Oxidative stress occurs when the accumulation of reactive oxygen species (ROS) in cells and tissues undermines the beneficial role of antioxidants within the body leading to the damage of lipids, DNA, RNA, and proteins, often resulting in cell death [68,95]. Increased ROS is suggested to play a pathogenic role in many central nervous system (CNS) diseases such as Alzheimer’s and Parkinson’s disease, as well as general ageing [95]. In IRDs, the role of ROS is unclear, but is possibly a contributing factor to accelerating cone degeneration [50,74].

In primary cone degeneration, ROS accumulation seemingly follows the activation of other cell death factors. Viringipurampeer et al. [71] showed that the formation of ROS was preceded by an increase in necroptotic RIP3 activity in cones in the *Pde6c^−/−^* zebrafish. Efforts to decrease RIP3 activity correlates with a decrease in ROS production [96], providing a potential therapeutic target. An increase in calpain activity was identified in chromatopsia mouse models and may also influence the increase in ROS production and accumulation [28,48]. Calpain cleaves apoptosis-inducing factor (AIF), an important mitochondrial molecule with redox activity; therefore, an increase in calpain activity may lead to mitochondrial dysfunction, and consequently, increased oxidative stress and increased ROS production [97,98]. Oxidative stress and ROS production proceed cell markers associated with cell death, but ROS may be too far downstream of a target for therapies in primary cone degeneration. However, targeting calpain activation or mitochondrial dysfunction directly could provide neuroprotection benefits and improve cone survival.

In contrast with primary cone degeneration, oxidative stress in secondary cone degeneration occurs due to the presence of hyperoxia-excess oxygen circulating in the retina that would otherwise have been taken up by the vast majority of rods that have now degenerated [50,51]. Oxygen levels in cones are therefore elevated, causing activation of nicotinamide adenine dinucleotide phosphate (NADPH) that influences ROS production [50,52]. Targeting ROS using antioxidants resulted in reduced cone cell death in retinitis pigmentosa mouse models [99,100]. Contrary to primary cone degeneration, oxidative stress and ROS production seemingly have an important upstream role in secondary cone degeneration, and could potentially be a therapeutic target for treating retinitis pigmentosa.

### 3.2. Endoplasmic Reticulum Stress

The endoplasmic reticulum (ER) is important for protein folding and lipid synthesis [101]. Pathological disruptions to the ER lead to cellular stress and the induction of cell death [102]. ER stress is implicated in both primary and secondary cone degenerations where death is more associated with apoptosis [29,103,104]. Interestingly, the only achromatopsia-associated gene not involved with the phototransduction cascade is the activating transcription factor 6 (*ATF6*), a gene important for regulating ER stress, and mutations in *ATF6* have been shown to lead to ER stress [105,106]. The question of why mutations in the *ATF6* gene have such a drastic effect on cone photoreceptor viability when ATF6 is expressed throughout the body is still not fully understood [104]. Retinal imaging of patients with ATF6 show not only foveal hypoplasia which is common in other achromatopsia mutations, but also absence of typical foveal development hallmarks. This led to theories that ATF6 is important for development of the fovea, where the majority of human cone photoreceptors reside [105].

As well as ATF6, other important ER stress proteins have been implicated in cone degeneration. Activation of ER stress causes a cascade effect which suppresses anti-apoptotic markers BCL2 and BCL-XL and promotes cell death through mechanisms such as apoptosis and autophagy (Figure 3). In achromatopsia mutations, excessive intracellular calcium (Ca^2+^) has been observed in cones, with Ca^2+^ toxicity being linked to cell death regulation [107,108,109,110]. This unregulated Ca^2+^ influx leads to ER stress indicated by the increase in ER stress markers such as 1,4,5-trisphosphate (IP_3_), eukaryotic initiation factor 2α (eIF2α), and CCAAT/-enhancer-binding protein homologous protein (CHOP) [28,29,110]. Increase in Ca^2+^ and ER stress is also evident in secondary cone degeneration models [47,111], but whether ER stress is the main contributor or just one aspect of secondary cone death is still unclear.

In a cone-dominant mouse model of primary cone degeneration used by Ma et al., the achromatopsia *Cnga3^−/−^* mouse was crossed with the *Nrl*^−/−^ mouse to produce a double knockout mouse with a rod-less retina. This allowed a focus on the effects of the *Cnga3^−/−^* mutation on cone photoreceptors and it was found that cones were degenerating through an ER stress-related caspase-dependent apoptosis. Protein kinase G (PKG) is a regulator of ER stress and has been shown to be upregulated in the *Cnga3^−/−^Nrl^−/−^* mouse [29] as well as retinitis pigmentosa models [112]. Induction of a PKG inhibitor to block an observed increase in PKG activity during primary cone degeneration reduced ER stress and prolonged cone survival in the *Cnga3^−/−^Nrl^−/−^* mouse [29]. PKG inhibition has also been studied in *Rd1* and *Rd2* mice to model its effects on photoreceptor degeneration, where PKG overactivity was shown to be harmful to both rod and cone photoreceptors [112]. Efficacy was determined by retinal thickness after injection with a PKG inhibitor at P9, and assessment at P16 showed that the treatment retained some of the retinal thickness in the treated compared with untreated *Rd1* mice. However, specific quantification of rod and cone photoreceptors numbers was not conducted. Further evaluation of PKG inhibition specifically in cone survival is required to evaluate the importance of PKG in ER stress in different IRD mouse models.

Targeting ER stress could potentially be an interesting treatment approach that could alleviate the detrimental effects of both primary and secondary cone loss. Indeed, ER stress has also been implicated in secondary cone degeneration. Leber congenital amaurosis (LCA) exhibits severe early-onset photoreceptor degeneration, and mutations have a drastic effect on cone photoreceptor viability [113]. The *Lrat^−/−^* mouse model of LCA14 undergoes severe cone degeneration due to S-opsin mislocalisation and aggregation, inducing ER stress in cone photoreceptors [114]. Administration of tauroursodeoxycholic acid (TUDCA), an ER chemical chaperone, alleviated ER stress in *Lrat^−/−^* and drastically slowed photoreceptors degeneration [115]. Application of TUDCA has been carried out in the *Rd1*, *Rd10* and *Rd16* models of retinitis pigmentosa. However, only the *Rd10* mouse showed significant improvements in photoreceptor structure and function, but this was only maintained for around 7 days post-treatment [116]. Therefore, the ER stress theory for cone degeneration may not be applicable to all IRD mutations.

### 3.3. Epigenetic Changes and Post-Transcriptional Regulation

Epigenetic changes are described as changes in the cell’s epigenome that influence gene expression, but do not fundamentally alter the genomic DNA sequence. In addition to methylation of promoter regions in DNA itself, the epigenome is also determined by the chromatin structure and accessibility of DNA to transcription factors for gene expression via histone modification. Examples of histone modification include acetylation that relaxes chromatin and allows genes to be turned “on” and methylation, which often but not always results in genes being turned “off” [117]. During cellular stress, chromatin modification takes place to recuperate a homeostatic state [118,119]. However, in IRDs, dysregulation of chromatin remodelling persists [37,120,121,122]. Treatments using proteins involved in chromatin remodelling, such as histone deacetylases (HDACs), have been tested in IRD mouse models, but their therapeutic mechanisms are only starting to be understood [121,122,123]. During normal development, HDACs are essential for photoreceptor differentiation. The inhibition of class 1 and 2 HDACs in healthy mice retinas from P2 to P10 resulted in decreased number of rod photoreceptors being developed, as well as a reduction in expression of rod differentiation factors [124]. In primary cone degeneration, excessive HDAC activity has been reported, and inhibition of HDACs provided short-term cone survival [121,123]. Excessive HDACs activation has also been shown to have a harmful effect in secondary cone degeneration, and again HDAC inhibition was shown to have an impact on preserving cone viability [120,121]. Although certain HDAC inhibitors have been FDA approved for cancer treatments [125], potentially fast-tracking approval for use in IRDs, off-target effects and longevity of these treatments need to be fully assessed within a retinal context.

Post-transcriptional regulation involves epigenetic changes at the RNA level and can be dysregulated in diseases, including the use of non-coding RNAs to modify gene expression such as micro-RNA (miRNA). miRNAs average about 22 nucleotides in length and attach to messenger RNA (mRNA) to block the translation into protein product [126]. The use of miRNA in cancer reduction has shown relative success [127,128,129,130], but no drugs have reached FDA approval yet. In the retina, miRNA are important for the maturation of photoreceptors [131,132,133]. The loss of a particular miRNA, miR-124, leads to decreased opsin expression and cone cell death [131]. The expression profiles of miRNAs in IRD retinas compared with healthy retinas have been examined in secondary [134,135], but not primary, cone degeneration models. Dicer is an important protein in the miRNA synthesis pathway, with conditional Dicer knockout mouse models created for both rods [136] and cones [132] to examine the importance of miRNA in each photoreceptor type. In the Dicer KO retina, rod degeneration occurred at P28 when rods were mature and postmitotic [136], whilst there was dysfunction of cone outer segments starting at P30 with a lack of degeneration [132]. This suggests that miRNAs are important for rod survival and cone functioning. Potential therapies could therefore be developed using miRNA technology targeting photoreceptors. Future research may also target particular dysregulated miRNAs in IRDs for novel therapies; however, miRNA research in the retina and in IRDs is still in its infancy, and identification of any dysregulated miRNAs in IRDs as well as their targets is still required.

## 4. Immunological Effects

The immune system works to maintain homeostasis and protect the body from negative effects of infection, cell degeneration, and physical trauma through eliciting an appropriate immune response [137]. During an immune response, certain immune cells become activated to overcome negative bodily conditions, depending on infection or disease type. Glial cells in the retina aid in the immune response and act to maintain homeostasis by providing support and protections of neurons [138,139]. However, in certain instances, over-reactive glial activation can lead to further damage to the affected area. This further degeneration can be seen in IRD models where photoreceptor death in the retina can be exacerbated by the activation of glial cells [56,57,140]. Treatment to circumvent the over-reactive glial activation may also therefore pose as a potential therapy option for IRDs.

### 4.1. Müller Glia

Müller glia are the predominant glial cells in the vertebrate retina, stretching radially across the retina. Their activation invokes a widespread pro-inflammatory response [141]. Increased Müller glia activity in mammalian retinal disease can contribute to degeneration in the retina, as these cells congregate to form a glial scar in the subretinal space, intensifying photoreceptor cell death [142,143]. In the primary cone degeneration *Cnga3^−/−^* mouse model, Müller glia activation is evident after eye opening [27] and had a reduced activation in response to treatments administered to preserve cone survival [29,144]. However, studies into other primary cone degeneration models are sparse. A study conductedby Hippert et al. [55] in four different retinitis pigmentosa mouse models described gliosis as being mutation-specific, but the extent of Müller glia activation did not correlate with disease severity. Targeting Müller glia in terms of suppression of gliosis to preserve cone survival in IRDs may be difficult due to disease variation.

In mammals, photoreceptors do not possess the ability to regenerate, meaning degeneration of photoreceptors due to disease or damage is irreversible [145]. Conversely, it has been demonstrated in the zebrafish and damaged chicken retina that Müller glia are able to undergo dedifferentiation into progenitor cells when undergoing active gliosis, and are then able to differentiate into photoreceptor cells, restoring their initial visual loss [146,147,148,149,150]. Induction of Müller glia differentiation into photoreceptors has been explored in murine models of retinal degeneration to varying degrees of success. Using human Müller glia with stem cell characteristics (hMSCs), in vitro proliferation of these cells using rod-specific differentiation factors and transplantation into the *Rho* P23H rat resulted in hMSC-derived photoreceptor cells migrating to the outer nuclear layer, where a significant increase in rod function was observed [151]. Transplantation using hematopoietic stem and progenitor cells (HSPCs) differentiated into retinal precursor cells, accompanied by Wnt activation, has also been successful in rescuing photoreceptor function in *Rd10* mice [152].

Expression of certain factors to induce Müller glia differentiation directly in vivo has also been explored in murine models [153,154]. Takeda et al. [153] showed that sub-toxic levels of glutamate stimulation induced Müller glia proliferation in healthy mice, allowing generation of photoreceptor cells without transplantation. In zebrafish, the *ASCL1* gene is upregulated in Müller glia after retinal damage and is important for regeneration, although this process is not present in mammals [148]. Induction of Müller glia *ASCL1* expression in retinal damaged mice initially leads to a neurogenic state. However, this neurogenic state in Müller glia is limited after P16 due to epigenetic HDAC regulation, meaning Müller glia lose their potential to proliferate. The increase in ASCL1 and inhibition of HDACs is associated with successful Müller glia differentiation and proliferation, restoring partial vision in N-methyl-D-aspartate (NMDA) damaged mouse retina [154]. Future research could explore how the results in damaged retinas in mouse models for Müller glia differentiation can be used in IRD mouse models for therapeutic translation into humans.

### 4.2. Microglia

Microglia are another glial cell type found in the retina that may influence retinal cell death through their phagocytic immune response [155]. Microglia act to clear debris and dying cells from the retina, but their excessive activation can lead to the release of harmful toxins and factors inducing cell death [57]. The invasion of microglia from the inner and outer plexiform layers into the photoreceptor layer during degeneration also influences the expression of neurotrophic factors, including nerve growth factor (NGF) [156]. Usually associated with cell survival, NGF can be produced by microglia to induce photoreceptor apoptosis via p75 receptors [157]. In IRD mouse models, microglia have been shown to contribute to the degeneration of photoreceptors through phagocytosis of non-degenerative photoreceptors [56,57]. In contrast, Sasahara et al. [158] reported that endogenous bone marrow (BM)-derived stem cells that differentiated into microglia protected photoreceptors in the *Rd1*, *Rd10*, and *Rd12* mouse models of retinitis pigmentosa by preventing vascular and neural degeneration. Consistent with this, inhibition of stromal-derived factor 1 (SDF-1), a chemokine that recruits BM-derived cells after injury, resulted in fewer BM-derived microglia and accelerated degenerative events in the retina [158]. Microglial manipulation through developing drugs for therapeutic benefits may be another example of a delicate homeostatic equilibrium that is hard to target due to potential detrimental effects of either its activation or inhibition.

In the secondary cone degeneration mouse models *Rd1* and *Rd10*, blocking the production of neurotoxic inflammatory factors produced by microglia via an AAV2/8-mediated delivery of the anti-inflammatory cytokine transforming growth factor beta type 1 (TGFBR1) showed a prolonged effect on cone survival [159]. However, the AAV2/8-TGFBR1 treatment had no effect on rod survival. Wang et al. [159] also confirmed that the treatment did not deplete microglia numbers; rather it had a downregulation effect on the inflammatory markers produced by microglia and had no observed side effects on vision. Furthermore, suppression of microglia demonstrated in retinitis pigmentosa animal models using tamoxifen, a drug used in breast cancer, mitigates photoreceptor cell death with relatively short-term effects [160,161]. Novel treatment options may be derived from these studies; however, suppression of the degenerative properties of microglia on photoreceptors is a short-term approach and would require a combination with a more substantial long-term treatment. Research is also limited on the effects of microglia on models of primary cone degeneration. Table 1 summarises mutation-independent treatments for IRDs that have been discussed in this review.

## 5. Future Perspectives

Despite specific death mechanisms not yet being identified, other forms of therapy are being investigated that target different cellular processes. Oxidative and endoplasmic reticulum stress are seemingly downstream processes that are a result of IRD mutations, and targeting these pathways may provide only short-term photoreceptor preservation effects [29,99,100,112,115,116]. Importantly, epigenetic changes using HDACs have already proved to be beneficial in combating cancer, and FDA-approved drugs are available [125]. This is relevant to IRDs because HDAC inhibition reduces photoreceptor degeneration in animal models [120,121,122,123]; thus, the use of such drugs could potentially fast-track clinical treatments to be used in IRDs. Emerging miRNA technology also has the potential to provide therapeutic benefit following its recent success in different types of cancer [127,128,129,130]. Manipulation of retinal glial cells may also prove valuable, as may the possibility of trans-differentiating existing Müller glia into photoreceptors. Rather than attempting to preserve remaining photoreceptors like other treatment options, the generation of functional photoreceptors to replace dead or dysfunctional photoreceptors in IRDs could potentially restore at least some level of vision [150,153,154].

## 6. Conclusions

Visual impairments are a growing problem in our current society and are accompanied by a large economic burden [3]. Many IRD patients have progressive degeneration over time, and life-long clinical monitoring and use of visual aids are required. Slowing the progression of photoreceptor degeneration in IRD patients with therapeutic treatments—especially ones aimed at the cone photoreceptors responsible for colour vision and visual acuity—are vital in preserving vision and improved quality of life in patients [7]. However, treatment options thus far have been unable to provide both efficacy and economic viability. This may be due to the complex mechanisms influencing cell death in IRD photoreceptors, particularly cones, having yet to be fully elucidated. This review highlights major cell death pathways and the conflicting literature pertaining to these pathways being found in primary and secondary cone degeneration IRDs, as well as distinguishing similarities and differences between primary and secondary cone degeneration. We also summarise associated cellular processes that occur due to IRD mutations and developments in treatments that target cone degeneration. To conclude, while primary and secondary cone degeneration may seemingly have different cell death mechanisms, the study and identification of common biological cell death markers in different IRD conditions has the potential to lead to an economically viable route to developing a common treatment approach that could be applied to multiple IRDs.

## Figures and Tables

**Figure 1 ijms-23-00726-f001:**
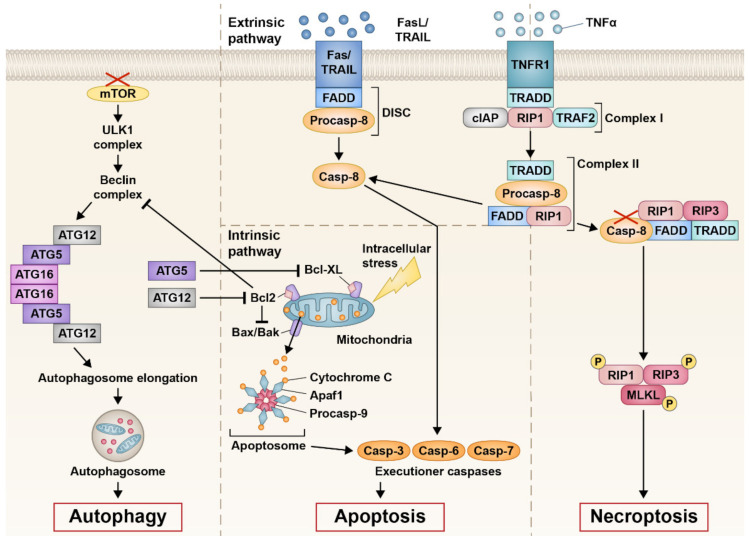
Interactions between the major pathways of cell death: autophagy, extrinsic apoptosis, intrinsic apoptosis, and necroptosis. Apoptosis can be initiated via binding of ligands FasL, TRAIL, or TNFα to their corresponding death receptor (extrinsic pathway) or via intracellular stress to the mitochondria (intrinsic pathway). Binding of FasL or TRAIL engages FADD and pro-caspase 8 to form the DISC complex, cleaving and activating caspase-8 to initiate downstream executioner caspase-3, -6, and -7. Activation of the proteolytic executioner caspases ultimately results in apoptosis. Binding of TNFα to its respective receptor, TNFR1, initiates formation of complex I (TRADD, cIAP, RIP1, TRAF2). TRADD and RIP1 recruit procaspase-8 and FADD to form complex II. Complex II can further go down the apoptosis pathway by activating caspase-8 or the necroptosis pathway by inhibiting caspase-8 when bound to RIP1, RIP3, FADD, and TRADD. Phosphorylation of RIP1, RIP3 and MLKL results in necroptosis. Intrinsic apoptosis can occur when there is an intracellular stressor causing mitochondrial dysfunctional, resulting in BAX/BAK activation and cytochrome c leakage when pro-survival markers BCL2 and BCL-XL are inactivated. Cytochrome c binds to APAF1 and procaspase-9 to form the apoptosome and downstream activation of executioner caspase to induce apoptosis. The autophagy process is initiated when there is a decrease in mTOR, causing activation of the Unc-51 like autophagy activating kinase 1 (ULK1) complex and further the Beclin complex. ATG5, -12, and -16 congregate to initiate the elongation process of autophagosome formation, inducing autophagy. Autophagy proteins interact with pro-survival proteins in that BCL2 blocks Beclin complex formation, whilst ATG12 blocks BCL2 and ATG5 blocks BCL-XL.

**Figure 2 ijms-23-00726-f002:**
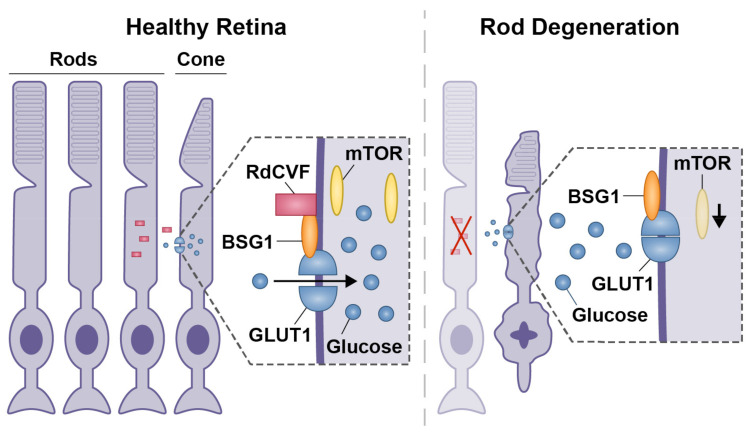
Theorised mechanism of secondary cone degeneration. In healthy retina, rods produce RdCVF to support cone function by binding to BSG1, allowing glucose to enter the cones via the GLUT1 receptor. When rods are degenerated as a result of disease, RdCVF is no longer produced and the cone GLUT1 receptor remains inactive. Dysregulation of glucose uptake into cones causes a decrease in mTOR, and cones begin to degenerate secondary to the loss of rods.

**Figure 3 ijms-23-00726-f003:**
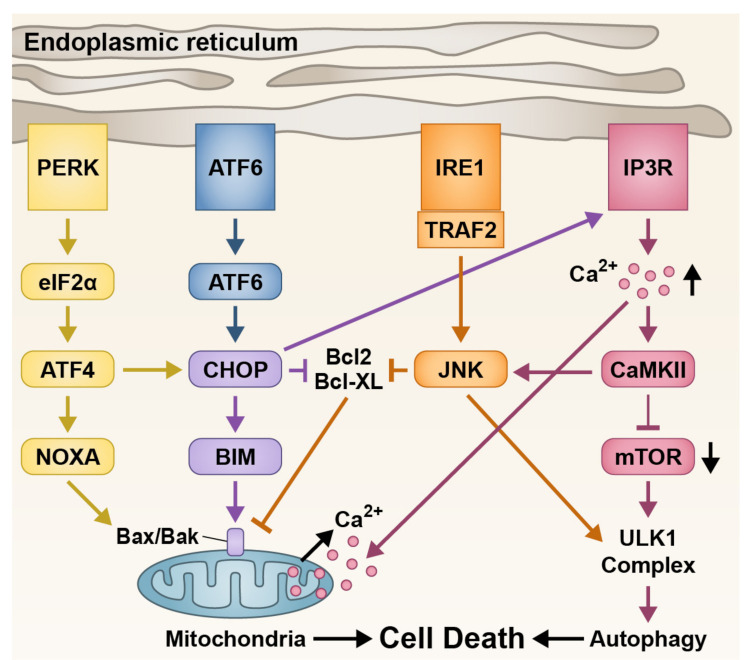
ER stress pathways through activation of transmembrane ER proteins to induce cell death. PERK and ATF6 triggers CHOP activation, blocking pro-survival proteins BCL2 and BCL-XL, and inducing mitochondria-mediated cell death. CHOP can also act on IP3R to release Ca^2+^ from the ER, leading to further mitochondrial dysfunction and autophagy-mediated cell death. IRE1 interacts with TRAF2 to activate the JNK pathway, blocking pro-survival proteins and increasing autophagy activity. ATF4: activating transcription factor 4; ATF6: activating transcription factor 6; BAK: Bcl-2 homologous antagonist killer; BAX: Bcl-2-associated X protein; BCL2: B-cell lymphoma-2; BCL-XL: B-cell lymphoma-extra large; BIM; Bcl-2-like protein 11; CaMKII: Ca^2+^/calmodulin-dependent protein kinase II; CHOP: CCAAT-enhancer-binding protein homologous protein; eIF2α: eukaryotic translation initiation factor 2α; IP3R: inositol 1,4,5-triphosphate receptor; IRE1: inositol-requiring enzyme 1; JNK: C-Jun N-terminal kinase; mTOR; mammalian target of rapamycin; NOXA: phorbol-12-myristate-13-acetated-induced protein 1, also known as (PMAIP1); PERK: protein kinase RNA-like endoplasmic reticulum kinase; TRAF2: TNFR1-associated factor 2; ULK1: Unc-51 like autophagy activating kinase 1.

**Table 1 ijms-23-00726-t001:** Potential mutation-independent treatments for IRDs which showed preserved cone and/or rod survival in animal models.

Cellular Process	Method	Model
Apoptosis	↓ Calpain	*Cnga3^−/−^* mouse [29] *Rd1* mouse [33]
Necroptosis	↓ RIP3	*Pde6c^−/−^* zebrafish [71] *Rd10* mouse [72] *Rho* S334ter rat [73]
Autophagy	↑ RdCVF	*Rho* P23H rat [86] *Rd1* and *Rho* P23H mice [87]
	↑ mTOR	*Pde6b^−/−^*, *Pde6g^−/−^*, *Rho^−/−^*, *Rho* P23H mice [54]
Oxidative Stress	↓ ROS	*Rd1* mouse [99,100]
Endoplasmic Reticulum Stress	↓ PKG	*Cnga3^−/−^Nrl^−/−^* mouse [29] *Rd1* and *Rd2* mice [112]
	TUDCA injection	*Lrat^−/−^* mouse [115] *Rd10* mouse [116]
Epigenetic Changes	↓ HDACs	*Cpfl1* mouse [121,123] *Rd1* mouse [120,122] *Rd10* mouse [122,123]
Immunological Changes	Müller glia differentiation	*Rho* P23H rat [150] Stimulation in healthy mice [153] NMDA retinal damaged mouse [154]
	Microglia inhibition	*Rd1* mouse [159] *Rd10* mouse [159,160] Mutant Mer tyrosine kinase associated retinitis pigmentosa (mutMerTK-RP) rat [161]

↓ Decrease in expression; ↑ Increase in expression

## Data Availability

Not applicable.

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
