# Peer review of "Primary and Secondary Cone Cell Death Mechanisms in Inherited Retinal Diseases and Potential Treatment Options"

_ijms, 2022, doi:10.3390/ijms23020726_

Round 1

Reviewer 1 Report

The paper by Brunet et al. describes the mechanisms underlying cone cell death in inherited retinal diseases (IRDs) and proposes possible treatment strategies, by providing an overview of in vivo evidence from the last two decades.

Despite Authors address an interesting topic in depth, they need to make some changes to improve the manuscript:

1) In line 70, it would be better to modify the title of section 2 to "Mechanisms of cone cell death".

2) In the final part of the introduction (lines 66-69), it would also be good to specify that, in order to meet the objective of the review, the most significant in vivo studies of the last 20 years concerning IRDs have been collected.

3) Authors could add a paragraph (following the introduction) which describes the most common forms of IRDs, and the related in vivo models developed, summarizing the latter ones in a sample table before discussing the mechanisms of cell death underlying these retinal diseases.

4) Authors should include a summary table including the suggested potential treatment options (i.e., using calpain inhibitors, targeting the RdCVF pathway, decreasing RIP3 activity or oxidative stress and ROS production, using miRNAs, through glia differentiation or suppression of microglia, etc.) before conclusions.

5) Section 5 appears to be a consideration rather than a conclusion. It should enclose what was reported in the cited studies and eventually indicate future prospects. The contribution of this review to scientific knowledge is also noteworthy.

6) Citations 13 (line 653) and 109 (line 829) are not complete, please correct them.

Author Response

The authors would like to thank the reviewer for suggestions as we agree with the reviewer and believe it has made improvements to our manuscript. We hope these changes are adequate for the reviewer.  

Please see the attachment for response to each reviewer's point. 

Reviewer 2 Report

I am glad of being part of the peer-review team of this paper. I think this review of the primary and secondary cone cell death mechanisms in retinal diseases will make an impact for the science also for the way you described how cell death mechanisms could join between them and make difficult to merge in one biological target for retinal diseases. Follow the minor changes Im sure you might have of the peer-review and this article would be published. I personally congratulate your team work and writing experience. 

Author Response

We would like to thank the reviewer for their positive feedback and hope to work with the reviewer in future endeavors.  

Round 2

Reviewer 1 Report

Authors have successfully made most of the suggested changes. However, still some point addressed can be improved:

1) It is not pleasant the expression "see Collin et al." in lines 81-82. Therefore, it is preferable to introduce the sentence as "Regarding a large animal IRD models, Collin et al. (17) and Winkley et al (18) offer a comprehensive overview on genes and associated mouse models involved in photoreceptor loss".

2) It is not necessary to add section 5 to include only Table 1. It would be better to first mention table 1 in the text, at the end of section 4 (i.e., "In Table 1 are summarized...") and then add it at the end of section 4.

3) Authors state that the conclusion section includes two paragraphs. The first paragraph highlights the contribution of the review to scientific knowledge and the second one the future prospects for treatment. However, although the conclusion section has been implemented, it consists of a single paragraph. Therefore, it would be better if the conclusions were separated from the future perspectives.
